# CAF-Associated Genes in Breast Cancer for Novel Therapeutic Strategies

**DOI:** 10.3390/biomedicines12091964

**Published:** 2024-08-29

**Authors:** Kanako Naito, Takafumi Sangai, Keishi Yamashita

**Affiliations:** 1Division of Advanced Surgical Oncology, Research and Development Center for New Medical Frontiers, Kitasato University School of Medicine, Sagamihara 252-0374, Japan; 2Department of Breast and Thyroid Surgery, Kitasato University School of Medicine, Sagamihara 252-0374, Japan; sangai@med.kitasato-u.ac.jp

**Keywords:** tumor microenvironment (TME), cancer-associated fibroblasts (CAFs), SPARC, breast cancer, colorectal cancer, triple-negative breast cancer

## Abstract

Breast cancer (BC) is the most common cancer in women, and therapeutic strategies for it are based on the molecular subtypes of luminal BC, HER2 BC, and triple-negative BC (TNBC) because each subtype harbors different unique genetic aberrations. Recently, features of the tumor microenvironment (TME), especially cancer-associated fibroblasts (CAFs), have been demonstrated to play a critical role in BC progression, and we would like to understand the molecular features of BC CAFs for novel therapeutic strategies. In a recent study, 115 CAF-associated genes (CAFGs) were identified in a public database of microdissection and microarray data (GSE35602) from 13 colorectal cancer (CRC) tumors. Using a public database (GSE10797) of 28 BC tumors, a similar analysis was performed. In BC, 59 genes from the 115 CAFGs identified in CRC (CRC CAFGs) were also closely associated with a CAFs marker, *SPARC* (R = 0.6 or beyond), and *POSTN* was of particular interest as one of the BC CAFGs with the highest expression levels and a close association with *SPARC* expression (R = 0.94) in the cancer stroma of BC tumors. In BC stroma, *POSTN* was followed in expression levels by *DKK3*, *MMP2*, *PDPN*, and *ACTA2*. Unexpectedly, *FAP* and *VIM* were not as highly associated with *SPARC* expression in the cancer stroma of BC tumors and exhibited low expression. These findings suggested that *ACTA2* might be the most relevant conventional CAFs marker in BC, and *ACTA2* was actually correlated in expression with many CRC CAFGs, such as *SPARC*. Surprisingly, the SE ratio values of the BC CAFGs were much lower (average SE = 3.8) than those of the CRC CAFGs (SE = 10 or beyond). We summarized the current understanding of BC CAFs from the literature. Finally, in triple-negative BC (TNBC) (n = 5), *SPARC* expression uniquely showed a close association with *COL11A1* and *TAGLN* expression, representing a myofibroblast (myCAFs) marker in the cancer stroma of the BC tumors, suggesting that myCAFs may be molecularly characterized by TNBC in contrast to other BC phenotypes. In summary, CAFs could have unique molecular characteristics in BC, and such TME uniqueness could be therapeutically targeted in BC.

## 1. Introduction

Breast cancer (BC) is the most common cancer, with more than 2,290,000 cases and 666,000 deaths each year worldwide [1], and its subtypes are defined as luminal BC, HER2 BC, and triple-negative BC (TNBC), classified according to therapeutic strategy [2,3,4]. Drug therapy for BC depends on the molecular subtype: endocrine therapy for luminal BC, anti-HER2 drugs for HER2 BC, and chemotherapy and immune-checkpoint inhibitors (ICIs) for TNBC.

In the BC project of TCGA (The Cancer Genome Atlas) [5], significant recurrent mutations were identified in *PI3K* (36%) and *TP53* (37%), while other genes were mutated at rates of less than 10%. *PI3K* mutations were found in 9% of TNBC, in contrast to other types (29~45%), while *TP53* mutations were seen in 12~29% of luminal BC, in contrast to HER2 BC (72%) and TNBC (80%). These findings suggested that the molecular etiology is different between luminal BC and HER2 BC or TNBC.

*PI3K* regulates signaling pathways important for cell growth, survival, and metabolism; overactivation of or abnormal mutations in *PI3K* can promote cell growth and cause resistance to treatment in BC [6], which can make it difficult for patients with certain *PI3K* mutations to respond to certain therapies. *TP53* is a tumor suppressor gene that detects and repairs cellular DNA damage or abnormalities or induces cell death (apoptosis) [7]. Mutations in *TP53* can impair the ability of cells to repair DNA and can increase cancer cell progression and resistance to therapy. Therefore, *TP53* mutations may play an important role as a predictor of therapeutic resistance.

*BRCA1* and *BRCA2* are genes involved in DNA repair, and mutations in both genes reduce the capacity for DNA double-strand break repair. BC patients with *BRCA1*/*2* mutations are known to be highly sensitive to PARP inhibitors, especially Olaparib [8]. This makes *BRCA1*/*2* mutations a potentially important predictor in treatment selection. Such genetic mutations may also affect the individual pathology and prognosis of BC. Therefore, the presence or absence of these mutations is an important factor to consider in the choice of treatment strategies and prognostic prediction.

Approximately 20% of TNBC tumors were found to have germline and/or somatic *BRCA1* or *BRCA2* mutations [5]. Cancer-associated fibroblast (CAFs) compositions change with BC progression, linking the ratio between S100A4(+) and PDPN(+) CAFs to clinical outcomes [9]. Their ratio is associated with disease outcomes across subtypes and is particularly correlated with *BRCA* mutations in TNBC, suggesting that CAFs subpopulations and the appropriate therapeutic strategies are different according to *BRCA* mutation status. Thus, molecular understanding of CAFs features is important for novel therapeutic strategies for BC.

## 2. CAFs Classification and Heterogeneity in BC

It is possible that diverse subpopulations of CAFs with different molecular features and functions may exist within the same tumor, which could have significant implications for therapy development. Recent molecular classification by single-cell transcriptomic analysis (scRNA technology) of an animal BC model clarified that CAFs were composed of vascular CAFs (vCAFs), matrix CAFs (mCAFs), cycling CAFs (cCAFs), and development CAFs (dCAFs) [10], of which the predominant group, vCAFs (~70%), may affect angiogenesis, while the second most abundant group, mCAFs (~20%), secrete extracellular matrix (ECM) materials such as *FBLN1*, *DCN*, *LUM*, and *FBLN2*, which were accompanied by *CDH1*, *PDGFRA*, and *LRRC15*. cCAFs showed high expression of Ki-67, representing proliferating CAFs; however, they accounted for only a minor CAFs subpopulation in BC.

In the primary study, signatures from vCAFs and mCAFs have demonstrated prognostic capabilities by correlating with metastatic dissemination in two large clinical cohorts involving over 2600 BC patients [10,11,12,13,14]. Additionally, the mCAFs signature showed a correlation with a stromal signature predictive of treatment response [15]. These findings collectively present an enhanced understanding of BC CAFs at the cellular, molecular, and functional levels, suggesting the potential for developing targeted drugs or biomarkers of clinical significance with improved precision [10].

CAFs can be alternatively classified according to their spatial distribution, where myofibroblasts (myCAFs) are located adjacent to tumor cells, in contrast to inflammatory CAFs (iCAFs), which are distributed distantly [16]. myCAFs may be greatly affected by secretion factors such as *TGFB1* from tumor cells through the *TGFB1*-*LRRC15* axis [17], and iCAFs can, in turn, be affected by myCAFs, at least through a process mediated by *IL1R1* downregulation by *TGFB1* [18].

Recently, CAF-associated genes (CAFGs) in colorectal cancer (CRC) were defined based on two criteria: (1) strong association (R ≥ 0.9, as indicated by underlined genes throughout this study) with expression of SPARC, a well-established stromal marker [19], in the cancer stroma of CRC tumors (GSE35607), and (2) specificity to the stroma (stroma/epithelia expression ratio, SE ratio ≥ 10) similar to SPARC (SE = 17.2) [20]. In CRC, CAFGs with the highest expression amounts (n = 115) included *FAP* (SE = 20.2), *ACTA2* (SE = 20.2), and *VIM* (SE = 17.8), the well-established CAFs markers [19]. These findings suggested that stromal expression of CAFGs may be well synchronized in CRC.

CAFGs were further classified into 3 categories by stroma specificity representing SE ratio (CAFGs of high SE ratio = 10 or beyond, semi-CAFGs of middle SE ratio = 5 or beyond and below 10, and L-CAFGs with low SE ratio = below 5) [20]. SE ratio showed stromal specificity because high expression amounts of the individual genes in tumor cells theoretically reduced values of the SE ratio; a high SE ratio may represent stroma-specific molecular status, while a low SE ratio would reflect their expression status in the tumor cells.

In this review article, CAFGs identified in CRC as a human cancer representative were initially exploited to those of BC for a literature search of CAFGs in BC (designated as BC CAFGs through this paper), because stromal components of different cancers are postulated to be shared across the human body, and they have been actually reported for their involvement in CAFs activation in BC. Subsequently, in silico analysis was conducted on data from the 28 micro-dissected BC tissues (GSE10797) [21], similar to those of CRC, in order to comprehend and summarize CAFGs in BC.

## 3. Material and Methods

### 3.1. Expression Profiles of the Microdissection Tissues of the 28 BC Tumors (GSE10797) 

The public database of cancer stromal expression after microdissection of the 28 BC tumors (GSE10797) was used in the microarray (Affymetrix HG U133A 2.0 GeneChip) harboring 22,277 genes.

### 3.2. Statistical Analysis

As a statistical method, the SE ratio was calculated as the ratio of stroma to epithelium after correcting the gene expression for GAPDH (probe: 217398_x_at AK026525) expression. Correlation coefficients between genes were calculated for the entire genes using the “CORRELATION” command in Excel software (Microsoft® Excel® for Microsoft 365 MSO (version 2407 build 16.0.17830.20166) 64-bit).

## 4. BC CAFGs and Current Understanding Its Functional Relevance

Among the 115 CAFGs identified in CRC (CRC CAFGs), 59 genes (64%) were also closely associated in expression with *SPARC* in cancer stroma of BC (R = 0.6 or beyond) from the 92 genes on the microarray chips, whereas their SE ratios were barely beyond 10 (average = 3.8, Appendix A and Figure 1a). Only 5 CRC CAFGs exhibited R-index with strSPARC = 0.9 or beyond in BC (gold letters in Figure 1b), in which *POSTN* (R = 0.94) was followed by *ASPN*, *COL1A2*, *COL5A2*, and *AEBP1* (Figure 1b). Recently, *SPARC* target genes (*COL3A1*, *COL5A1*, *COL1A1*, and *C1S*) were identified through *SPARC* knockdown in YS1 cells (a CAFs cell line, and these genes similarly showed close association with *SPARC* in BC stroma, as depicted in Figure 1c.

CAFGs with the highest expression amounts in GSE35602 (CRC) [22] have been literally described in the context of BC CAFs. In CRC, *SPARC* showed the highest expression (Appendix A), followed by *VIM*, *COL12A1* (SE = 18.3), *DKK3* (SE = 13.9), *ACTA2*, *INHBA* (SE = 20.9), *ANTXR1* (SE = 24.3), *PDGFRB* (SE = 14.8), *FAP*, *SLC24A3* (SE = 23.6), *ITGA11* (SE = 16.4), *MMP2* (SE = 18.4), *CCL2* (SE = 18.0), *POSTN* (SE = 15.4), *PDPN* (SE = 11.2), and *LRRC15* (SE = 10.5).

On the other hand, expression amounts of BC were different from those of CRC (Figure 2a, upper panel), where *POSTN* (breast SE, bSE = 3.6) showed the highest expression in BC tumors (GMS272720) differently from CRC, followed by *SPARC* (bSE = 3.5), *DKK3* (bSE = 4.1), *MMP2* (bSE = 4.9), *ACTA2* (bSE = 1.7), *PDPN* (bSE = 5.4), *FAP* (bSE = 4.3), *PDGFRB* (bSE = 4.6), *LRRC15* (bSE = 3.4), *ANTXR1 * (bSE = 2.8), *VIM* (bSE = 2.7), *SLC24A3* (bSE = 0.9), *INHBA* (bSE = 2.3), and *CCL2* (bSE = 5.0).

Surprisingly, SE ratios of CAFGs in BC were barely beyond 10 in contrast to those in CRC (CRC CAFGs were defined as SE ratio = 10 or beyond), and stromal specificity of CAFGs in BC was uniquely lower in contrast to that in CRC.

POSTN-positive stromal cells were recently demonstrated to guide lymphovascular invasion by cancer cells in BC [23]. POSTN-expressing CAFs were spatially found at periductal and perivascular margins and were enriched at lymphatic vessel peripheries, a highly metastatic cancer cells mobilize POSTN-expressing CAFs in the primary tumor site that promote collagen remodeling and collective cell invasion within lymphatic vessels and ultimately to sentinel lymph nodes. Consistent with this report, *POSTN* expression was closely correlated with *SPARC* expression as well as many collagen (COL) family genes in the cancer stroma of the BC tissues (Figure 1b,c).

Hypoxia inhibits prolyl hydroxylase domain protein 2 (PHD2), leading to hypoxia-inducible factor-1A (HIF1A, bSE = 1.8) stabilization, reduced expression of *ACTA2* and *POSTN* in BC [24]. Loss of PHD2 in CAFs phenocopies the effects of hypoxia, which can be prevented by simultaneous depletion of HIF1A. Treatment with the PHD inhibitor DMOG in an orthotopic BC model significantly decreases spontaneous metastases to the lungs and liver, which is associated with decreased stiffness and fibroblast activation. PHD2 depletion in CAFs co-injected with tumor cells similarly prevents CAFs-induced metastasis to the lungs and liver. These findings suggested POSTN- and/or ACTA2-expressing CAFs (myCAFs) are more aggressive than HIF1A-expressing CAFs (putatively representing iCAFs) in BC (Figure 3a).

*VIM*, *ACTA2*, and *FAP* have been considered as CAFs markers even in BC [25]. scRNA analysis clarified that CAFs are classified into vCAFs, mCAFs, cCAFs, and dCAFs in the spontaneous BC mice model [10], and the three CAFs markers, as well as SPARC, were expressed in all four subtypes of CAFs, hence designating them as panCAF markers (Figure 3b). In our in silico data, only *ACTA2* expression was closely associated (R = 0.6 or beyond) with *SPARC* expression in BC (Figure 1a). More importantly, *VIM* and *FAP* expressions were uniquely low (Figure 2a, upper panel) in contrast to CRC (Appendix A). *ACTA2* is therefore molecularly characterized by CAFs activation and is relevant to BC differently from *VIM* and *FAP*. Actually, close association with *ACTA2* expression was commonly shared with BC CAFGs (Figure 3c). Gene probes closely associated with *VIM* (R = 0.8 or beyond) in cancer stroma of the BC tumors were few (0) in contrast to *FAP* (646) and *ACTA2* (214).

Surprisingly, stromal fibroblasts from surgical margins (TCFs) expressed high levels of ACTA2 and CXCL12 and exhibited higher migratory/invasiveness abilities [26]. Indirect co-culture showed that TCF cells enhance the proliferation of non-cancerous mammary epithelial cells and the epithelial-to-mesenchymal transition (EMT) of BC cells. Moreover, TCF and CAFs cells increased the level of PCNA, MMP-2 (bSE = 4.9), and the phosphorylated/activated form of AKT (*AKT1* and *AKT3* were L-CAFGs in CRC). *MMP2* expression was uniquely high in BC (Figure 2a, top panel).

FAP delineates heterogeneous and functionally divergent stromal cells in immune-excluded BC. Mouse and human BCs were used to examine stromal cells expressing FAP, a deemed surface marker for CAFs [27]. Two discrete populations of FAP-positive mesenchymal cells were identified on the basis of podoplanin (PDPN) expression: FAP + PDPN + CAFs suppressed the proliferation of T cells in a nitric oxide-dependent manner, whereas FAP + PDPN pericytes were not immunosuppressive. Furthermore, a much higher expression amount of *PDPN* was seen compared to that of *FAP* in BC (Figure 2a, top panel).

On the other hand, circulating CAFs were identified in BC by the combination of CAFs markers, FAP, and ACTA2 co-expression (cCAFs) [27]. In patients with BC, the presence of cCAFs was detected in 30/34 (88%) patients with metastatic disease and 3/13 (23%) patients with localized BC with long-term disease-free survival, whereas no cCAFs were detected in healthy donors. Thus, the presence of cCAFs may be associated with clinical metastasis, suggesting that cCAFs may complement circulating tumor cells (CTCs) as a clinically relevant biomarker in metastatic BC.

Stromal expression of *Dickkopf-3* (*DKK3*) was also associated with aggressive BC [28], and *DKK3* expression is relatively high in BC (Figure 2a). *DKK3* is a heat shock factor 1 (HSF1) effector that modulates the pro-tumorigenic behavior of CAFs. In BC, *DKK3* orchestrates a concomitant activation of β-catenin and Yes-associated Protein 1 (YAP)/TAZ, where β-catenin is dispensable for CAFs-mediated ECM remodeling, cancer cell growth, and invasion, but DKK3-driven YAP/TAZ activation is required to induce tumor-promoting phenotypes.

*Integrin α11* (*ITGA11*)/*PDGFRB* were identified as CRC CAFGs (Appendix A). In BC, ITGA11/PDGFRB + CAFs displayed tumor-promoting features [29], though their expressions in BC may be relatively small and weak association with strSPARC (Figure 2a). In the preclinical MMTV-PyMT mouse model, *ITGA11* deficiency led to a drastic reduction of tumor progression and metastasis. Mechanistically, ITGA11 pro-invasive activity relies on its ability to interact with PDGFRB in a ligand-dependent manner. Pharmacological inhibition of PDGFRB impaired tumor cell invasion induced by ITGA11-positive CAFs.

Residual ANTXR1 + myCAFs after chemotherapy inhibited anti-tumor immunity via the YAP1 signaling pathway [30]. The content in the ECM-producing ANTXR1 + CAFs cluster (ECM-myCAFs) is the most affected by chemotherapy. Moreover, functional assays demonstrated that ECM-myCAFs reduce CD8 + T-cell cytotoxicity through a YAP1-dependent mechanism, suggesting an important role of YAP1 in myCAFs of BC.

myCAFs are characterized by ECM proteins and TGFβ signaling and are also indicative of primary resistance to immunotherapies [31]. ECM-myCAFs upregulate PD-1 and CTLA4 protein levels in regulatory T lymphocytes (Tregs), which, in turn, increases TGFβ-myCAFs cellular content. Thus, these data highlight a positive feedback loop between specific myCAFs clusters and Tregs and uncover their role in immunotherapy resistance.

RNA was extracted from low-passage cultures of CAFs and normal fibroblasts (NFs) and analyzed with Affymetrix Human Genome U133 Plus 2.0 arrays [32]. Twenty-one genes (27 probe sets) were upregulated in CAFs, as compared with NFs, which included *ST6GALNAC5* (bSE = 2.4), *SLC24A3* (bSE = 0.9), and *WISP1* (SE = 2.3) upregulated in BC CAFs. Known functions of these genes relate to paracrine or intracellular signaling, transcriptional regulation, ECM, and cell adhesion/migration. Nevertheless, *SLC24A3* showed low SE (<1) in our in silico data of BC. 

Adrenergic-mediated increases in *INHBA* drive CAFs phenotypes [33]. Daily restraint stress resulted in significantly increased CAFs activation and was abrogated by a nonspecific β-blocker. Adrenergic signaling-induced CAFs had significantly higher levels of collagen than control tumors. BC-secreted factors promote lung metastasis by signaling systemically to induce a fibrotic premetastatic niche, and *INHBA* expression in human patients with BC was associated with lung metastatic relapse and poor survival [34], although the molecular mechanism of *INHBA* contribution to metastasis remains elusive at present.

CAFs activated by cocultured BC cells produce higher levels of chemokine (C-C motif) ligand 2 (CCL2, bSE = 5.0), which stimulates the stem cell-specific, sphere-forming phenotype in BC cells and cancer stem cell (CSC) self-renewal [35]. Increased *CCL2* expression in activated fibroblasts required STAT3 activation by diverse BC-secreted cytokines, and in turn, induced *NOTCH1* expression and the CSC features in BC cells, constituting a cancer–stroma–cancer signaling circuit. In a xenograft model of paired fibroblasts and BC tumor cells, the loss of *CCL2* significantly inhibited tumorigenesis and *NOTCH1* expression.

Recently, collagen type XII (*COL12A1*) was newly identified as a critical component that regulates collagen type I organization [36], although our in silico analysis did not include *COL12A1* among BC CAFGs (Figure 1a) because the R-index with *SPARC* was below 0.6. Proteomics with single-cell transcriptomics and genetic manipulation model clarified that CAFs-secreted collagen XII alters collagen I organization to create a pro-invasive microenvironment supporting metastatic dissemination of BC [36].

## 5. CAFGs Collagens in BC and Novel Therapeutic Potential

Fibroblast-like cells are termed adipocyte-derived fibroblasts (ADFs) adjacent to breast tissues. These cells exhibit enhanced secretion of fibronectin (FN1, bSE = 8.2) and collagen I (COL1A1, bSE = 3.3), increased migratory/invasive abilities, and increased expression of the CAFs marker, *fibroblastic-specific protein 1*, *FSP1* (*S100A4,* bSE = 2.6) but not *ACTA2* [37]. Generation of the ADF phenotype depends on the reactivation of the Wnt/β-catenin pathway in response to WNT3A. Thus, control of ADFs may have a therapeutic potential in BC (Figure 4).

Impairment of a distinct CAFs population limits tumor growth and metastasis [38], where genetic deletion of the Endo180 (MRC2, bSE = 4.3) receptor, predominantly expressed by a population of matrix-remodeling CAFs, profoundly limits tumor growth and metastasis. In *MRC2* deletion mice, a remarkable reduction of collagen deposition was detected by Masson Trichrome staining. *MRC2*-induced ECM remodeling including dysregulation of collagen components could be a promising therapeutic target of BC (Figure 4).

CAFs require proline synthesis by PYCR1 (bSE = 2.0) for the deposition of pro-tumorigenic ECM. PYCR1 is a key enzyme for proline synthesis and is highly expressed in the stroma of BC patients and CAFs [39]. Reducing PYCR1 levels in CAFs is sufficient to reduce tumor collagen production, tumor growth and metastatic spread in vivo, and cancer cell proliferation in vitro. Both collagen and glutamine-derived proline synthesis in CAFs are epigenetically upregulated by increased pyruvate dehydrogenase-derived acetyl-CoA levels. PYCR1 is a cancer cell vulnerability and a potential target for therapy. Therefore, this finding provides evidence that targeting PYCR1 may have the additional benefit of halting the production of a pro-tumorigenic ECM (Figure 4).

## 6. Semi-CAFGs in BC and Novel Therapeutic Potential

Semi-CAFGs in CRC were defined as SE ratio = 5 or beyond plus below 10 with R-index with *SPARC* of 0.9 or beyond (Appendix A), and such genes have also been reported in BC like *TIMP3*, *RARRES2*, *TIMP2*, *NOTCH4*, *IGFBP6*, *ST6GALNAC5*, and *TGFB1.* Among them, *SPARC* expression was closely (R = 0.6 or beyond) correlated with *TIMP3, WISP1,* and *TGFB1* in BC (Figure 5a). Interestingly, *TIMP3* showed the highest expression, like CRC (Appendix A).

Intriguingly, exosomes produced by TIMPless fibroblasts induce cancer cell motility and cancer stem cell markers [40]. The proteome of these exosomes is enriched in ECMs and metalloproteinase, *ADAM10* (bSE = 2.3). Exosomal *ADAM10* increases *aldehyde dehydrogenase* expression through Notch receptor activation and enhances motility through the GTPase RHOA in BC cells. Intriguingly, *ADAM10* knockdown in TIMPless fibroblasts abrogates their CAFs function.

Stimulation of non-metastasis-promoting normal fibroblasts with TGF-B (*TGFB1*, SE = 2.3), *FGF-2*, *HGF*, and *PDGF-B* led to the acquisition of their metastatic capacity [41]. Resident human mammary fibroblasts progressively convert into myCAFs over the course of tumor progression. These cells increasingly acquire two autocrine signaling loops, mediated by *TGFB1* and *CXCL12* cytokines, which both act in autostimulatory and cross-communicating fashions. These autocrine-signaling loops initiate and maintain the differentiation of fibroblasts into myCAFs and the concurrent tumor-promoting phenotype. Collectively, these findings indicate that the establishment of the self-sustaining *TGFB1* and *CXCL12* autocrine signaling gives rise to tumor-promoting CAFs (myCAFs) [42].

The stromal fibroblasts isolated from BC tissues showed CAFs characteristics with high expression levels of *ACTA2* and *CXCL12*. The CAFs-conditioned medium (CM) transformed BC cell lines into more phenotypes, including enhanced ECM adhesion, migration, and invasion, and promoted epithelial–mesenchymal transition (EMT). CAFs secreted more TGFB1 than TGFB2 (SE = 1.1) and TGFB3 (SE = 2.4) and activated the TGFB/SMAD signaling pathway in BC cells. The EMT phenotype of BC cells induced by CAFs-CM was reversed by blocking *TGFB1* signaling [43].

The transcriptional regulator heat shock factor 1 (HSF1, SE = 2.2) is frequently activated in CAFs, where it is a potent enabler of malignancy. HSF1 drives a transcriptional program in CAFs that complements, yet is completely different from, the program it drives in adjacent cancer cells. This CAFs program is uniquely structured to support malignancy in a non-cell-autonomous way. Two central stromal signaling molecules—*TGFB1* and *CXCL12*—have a critical role [44].

Angiotensin (ANG1-7) is an endogenous peptide hormone of the renin–angiotensin system that has antiproliferative properties. In CAFs isolated from orthotopic BC, the Angiotensin hepta-(7)-peptide markedly attenuated in vitro growth as well as reduced FN1, TGFB1, and ERK1/ERK2. An associated increase in the MAPK phosphatase DUSP1 following treatment with ANG1-7 suggested a potential mechanism by which the heptapeptide reduced MAPK signaling. Consistent with these in vitro observations, immunohistochemical analysis of ANG1-7-treated orthotopic BC revealed reduced *TGFB1* and increased *DUSP1* [45]. Selective angiotensin blockers alleviate immunosuppression and improve CTL activity, enabling dramatically improved responses to immune-checkpoint blockers in mice with primary as well as metastatic BC [46,47].

CAFs increase the stiffness of the ECM and promote anisotropic fiber orientation, two mechanical signals generated through a SNAIL1-/RHOA- (SE = 0.6)/ACTA2-dependent mechanism that sustains oriented tumor cell migration and invasiveness. SNAIL1-depleted CAFs failed to acquire myCAFs traits in response to TGFB1, including RHOA activation, ACTA2-positive stress fibers, increased FN1 fibrillogenesis, and production of a stiff ECM with oriented fibers [48,49]. These findings suggested that *SNAIL1* (bSE = 2.5) is critical for *TGFB1*-induced ss. Intriguingly, CCL18 signaling from tumor-associated macrophages (TAMs) activates fibroblasts to adopt a chemoresistance-inducing phenotype [50].

*TGFB1* is known to be a critical mediator of the CAFs, and osteopontin (OPN/SPP1, bSE = 2.4) expression in tumors is associated with more aggressive phenotypes and poor patient outcomes. Tumor-derived SPP1 engenders mesenchymal stem cells (MSCs)-to-CAFs transformation in the TME to promote tumor growth and metastasis via the SPP1-MZF1-TGFB1 pathway [51]. SPP1 was sufficient to induce fibroblast reprogramming and neutralizing antibodies against SPP1-blocked fibroblast activation induced by tumor cells. The ability of secreted SPP1 to activate mammary CAFs relied upon its known receptors CD44 and αVβ3 integrin [52].

*WNT7A* (bSE = 2.9) is a key factor secreted exclusively by aggressive BC cells, which induces CAFs conversion. *WNT7A*-mediated fibroblast activation is not dependent on classical Wnt signaling. Instead, *WNT7A* potentiates TGFB receptor signaling [53]. On the other hand, *ZNF32* (bSE = 1.6) prevents the activation of CAFs through negative regulation of *TGFB1* transcription in BC [54].

CAFs subpopulations in metastatic lymph nodes (LNs) were myCAFs subsets, CAFs-S1 and CAFs-S4, which accumulate in LNs and correlate with cancer cell invasion [55]. By developing functional assays on primary cultures, these subsets promoted metastasis through distinct functions. While CAFs-S1 stimulate cancer cell migration and initiate an EMT through *CXCL12* and TGFB pathways, highly contractile CAFs-S4 induce cancer cell invasion in 3 dimensions via NOTCH signaling (*NOTCH1*, bSE = 3.0; *NOTCH2*, bSE = 2.6; *NOTCH3*, bSE = 1.4; *NOTCH4*, SE = 3.0).

Recently, lipid-associated macrophages (LAMs) have been induced by CAFs, and at the single-cell level a monocyte-derived STAB1 (bSE = 3.3) + TREM2 (bSE = 2.5) high LAM subpopulation with immune suppressive capacities that is expanded in patients resistant to the immune-checkpoint blockade. Genetic depletion of this LAM subset in mice suppressed TNBC tumor growth. Cell-to-cell interaction modeling and assays in vitro demonstrated the role of the inflammatory CXCL12-CXCR4 axis in CAFs-myeloid cell crosstalk and recruitment of monocytes in tumor sites [56].

Recently, *TGFB1* and the Janus kinase signaling cascades were validated as actionable targets to counteract the CAFs-induced modulation of estrogen receptor (ER) activity. Finally, genes that were downregulated in cancer cells by CAFs were predictive of poor response to endocrine treatment, proposing further novel targets to disrupt the crosstalk between CAFs and tumor cells to reinstate treatment response to endocrine therapy in BC patients [57].

The strongest two-cell circuit motif was isolated by culturing fibroblasts and macrophages in vitro and analyzing their dynamics and transcriptomes. This isolated circuit recapitulates the hierarchy of in vivo interactions and enables testing the effect of ligand–receptor interactions on cell dynamics and function, as demonstrated by identifying a mediator of CAFs–TAMs interactions—*RARRES2* (bSE = 2.5), and its receptor *CMKLR1* [58].

Insulin-like growth factor binding protein 6 (IGFBP6, bSE = 2.9) increased insulin-like growth factor-1 (IGF1, SE = 6.7) in the CM, where IGFs were investigated as key chemotactic factors [54]. Silencing *IGFBP6* or *IGF1* expression in epithelial cells or blocking *insulin-like growth factor 1 receptor *(*IGF1R*, bSE = 2.7) activity on fibroblasts significantly altered fibroblast mobilization. Stromal fibroblast transition to CAFs is linked through the IGFs/IGF1R axis in BC [59].

Metastatic BC-associated CAFs have enhanced pro-tumorigenic properties related to increased *IGF2* expression. CAFs from different metastatic sites (mCAFs) and compared with CAFs from primary tumors increased expression of IFN and IGF2 in the former. Cluster analysis revealed two groups of mCAFs, with the liver mCAFs clustering together, with increased *PDGFA* expression [60].

## 7. CAFGs with Low SE (L-CAFGs) and Novel Therapeutic Potential in BC

Although the SE ratio was low (below 5), many genes were closely (R = 0.9 or beyond) associated in expression with *SPARC* in GSE35602, and they could be designated as CAFGs with low SE (L-CAFGs) in CRC and have been reported their involvement in CAFs activation (Appendix A). The L-CAFGs according to expression amounts are shown in Appendix A, ranked as top of the L-CAFGs in BC like *ATF4*, followed by *ITGAV*, *TGFBR2*, *NOTCH2*, *TIMP1*, *CRELD2*, *HIF1A*, *MAPK14* (*p38*), *SMAD2*, *ROCK1*, *ADAM10*, *NOTCH1*, *MAPK1* (*ERK2*), *IGF1*, *MRC2*, *NFKB2*, *AKT1*, *AKT3*, and *IGF1R*.

Among them, *SPARC* expression was closely (R = 0.6 or beyond) correlated with the following genes according to R-index in BC: *BNP3L,* followed by *ITGAV*, *NFKB2*, *AKT3*, *ADAM10*, *IGF1R*, *MAPK14 *(p38), *MAPK1 *(Erk2), *AKT1*, *CRELD2*, *TGFBR2*, and *NOTCH2* (Figure 5c). Interestingly, *MAPK14 *(p38) and *ITGAV* showed the highest expression differently from CRC.

Tumor-derived osteopontin (OPN/SPP1) drives the resident fibroblasts to myCAFs. Tumor-derived SPP1 achieves this transition by engaging CD44 (bSE = 2.5) and ITGAV (bSE = 2.2)/ITGB3 (bSE = 2.7) on the fibroblast surface, which mediates signaling via AKT (AKT1 and AKT3) and ERK (ERK2 = MAPK1) to induce TWIST1 (bSE = 2.5)-dependent gene expression in BC [61]. The OPN-driven CAFs then secrete CXCL12, which in turn triggers EMT in the tumor cells. AKT (AKT1/AKT3) substrate Girdin (KIAA1212), an actin-binding protein that regulates cell migration, is expressed and activated by AKT phosphorylation in CAFs [62].

CD10 (membrane metallopeptidase, MME) and GPR77, a C5a receptor, specifically define a CAFs subset correlated with chemoresistance and poor survival in multiple cohorts of BC patients [63]. CD10 + GPR77 + CAFs promote tumor formation and chemoresistance by providing a survival niche for cancer stem cells (CSCs). Mechanistically, CD10 + GPR77 + CAFs are driven by persistent NF-κB (NFKB2, bSE = 3.2) activation via p65 phosphorylation and acetylation, which is maintained by complement signaling via GPR77.

Disruption of prostaglandin E2 (PGE2) signaling in CAFs limits BC growth but promotes metastasis. PGE2 plays a paradoxical role in CAFs activation and tumor progression [64]. Restricting PGE2 signaling via knockout of microsomal prostaglandin E synthase-1 (PTGES, bSE = 0.8) in PyMT mice or of the prostanoid E receptor 3 (EP3) in CAFs suppressed BC growth associated with strong CAFs proliferation. CAFs proliferation upon EP3 inhibition required p38 MAPK (MAPK14) signaling. Mechanistically, TGFB1-activated kinase-like protein (TAK1, C21ORF7, bSE = 3.3) was identified as a negative regulator of p38 MAPK activation.

ROCK (ROCK1, bSE = 4.5)-mediated selective activation of PERK signaling causes fibroblast reprogramming and tumor progression through a CRELD2 (bSE = 0.7)-dependent mechanism. PERK-mediated CAFs education was performed downstream of ROCK, and *CRELD2* is regulated by PERK-regulated ATF4 (bSE = 1.5). Depleting *CRELD2* suppressed tumor progression, demonstrating that the paracrine ROCK-PERK-ATF4-CRELD2 axis promotes the progression of BC, with implications for cancer therapy [65].

On the other hand, CAFs-specific *TGFBR2* (bSE = 2.2) expression correlated with improved recurrence-free survival, and multivariate analysis confirmed CAFs-TGFBR2, but not CAFs-SMAD2 (SE = 1.5) phosphorylation, to be an independent prognostic marker. Knockdown of *TGFBR2* in CAFs resulted in increased cell growth, proliferation, and clonogenic survival, suggesting that TGFB pathway activation is not necessarily associated with BC tumor progression, but reflected negative feedback.

Average bSE ratios of semi-CAFGs and L-CAFGs classified in CRC (BC semi-CAFGs and BC L-CAFGs, with R-index with strSPARC in BC = 0.6 or beyond) were 2.6 and 2.6, respectively, and they were not so different from those of CAFGs (average = 3.8), unlike CRC. These findings suggested that CAFGs cannot be clearly delineated into CAFGs, semi-CAFGs, and L-CAFGs by bSE ratios in BC. This difference may be due to differences according to cancer species or might be due to different assays (microarrays) used for analysis (Agilent microarray in CRC and Affymetrix microarray in BC, or 13 cases in CRC and 28 cases in BC, respectively).

Finally, we explored CAFGs in triple-negative BC (TNBC), although they were only in four cases. Among the BC CAFGs with R-index with strSPARC = 0.9 or beyond, *AEBP1*, *COL5A2*, *POSTN*, and *COL1A2* again exhibited high R-index = 0.9 or beyond (Figure 5d). On the other hand, the additional 5 genes showed a high R-index = 0.9 or beyond (Figure 5e), and *COL11A1* and *TAGLN* expressions as shown in red letters were uniquely associated with *SPARC* expression in cancer stroma of the TNBC tumors (both R-index below 0.6 in total BC). The highest expression amounts of TNBC are shown in Figure 1b,c. As *COL11A1* is a myCAFs marker [66], TNBC may be molecularly characterized by myCAFs accumulation in contrast to other BC.

Limitations of this study include the small sample size of the data analyzed, especially for TNBC. The data used in this study may not be representative of the broader population of BC patients, and further studies using larger datasets and different patient populations are needed to validate the findings regarding the BC relevance of *ACTA2*. Moreover, the greatest limitation of this study is that it relied exclusively on gene expression data to identify CAFs markers and did not authenticate the findings through other methodologies such as functional assays or protein expression, given that this paper is a kind of review article.

## 8. Conclusions

Recent studies clarified a future perspective in which TME shows a hierarchy of cell–cell interactions dominated by CAFs [58], where Mayer S et al. provided an experimental–mathematical approach to decompose the TME into specific small circuits of interacting cell types. They herein discovered, using scRNAseq data, a hierarchical network of interactions, with CAFs at the top secreting factors primarily to TAMs, and actually identified a mediator of CAFs–TAMs interactions. The complexity of the TME may be thus simplified by identifying such small circuits, facilitating the development of novel strategies to modulate the TME.

This review article summarizes the molecular features of CAFs in BC compared to CRC and identifies the unique molecular features of CAFs in BC. CAFGs classified in CRC have also been reported as their relevance in BC CAFs; however, they were not so clearly delineated according to SE ratio differently from CRC, because CRC showed a much higher SE ratio than in BC. This may propose a molecular difference between BC and CRC, reflecting differential biology and clinical outcomes. In BC, *SPARC*-defined CAFGs (R = 0.6 or beyond) only included *ACTA2* as a CAFs marker instead of *FAP* and *VIM*. Moreover, among BC CAFGs, *POSTN* expression was found to be the highest, followed by *DKK3* and *MMP2*. In BC, POSTN-positive stromal cells were recently demonstrated to guide lymphovascular invasion by cancer cells [23], and *DKK3* was demonstrated to orchestrate activation of YAP/TAZ, which is required to induce tumor-promoting phenotypes. Quadruple TIMP knockout (TIMPless) fibroblasts to unleash MMP activity and complete Timp loss is sufficient for the acquisition of hallmark CAFs functions in BC, which was mediated by exosome *ADAM10* [40]. These findings might propose unique therapeutic points in BC.

## Figures and Tables

**Figure 1 biomedicines-12-01964-f001:**
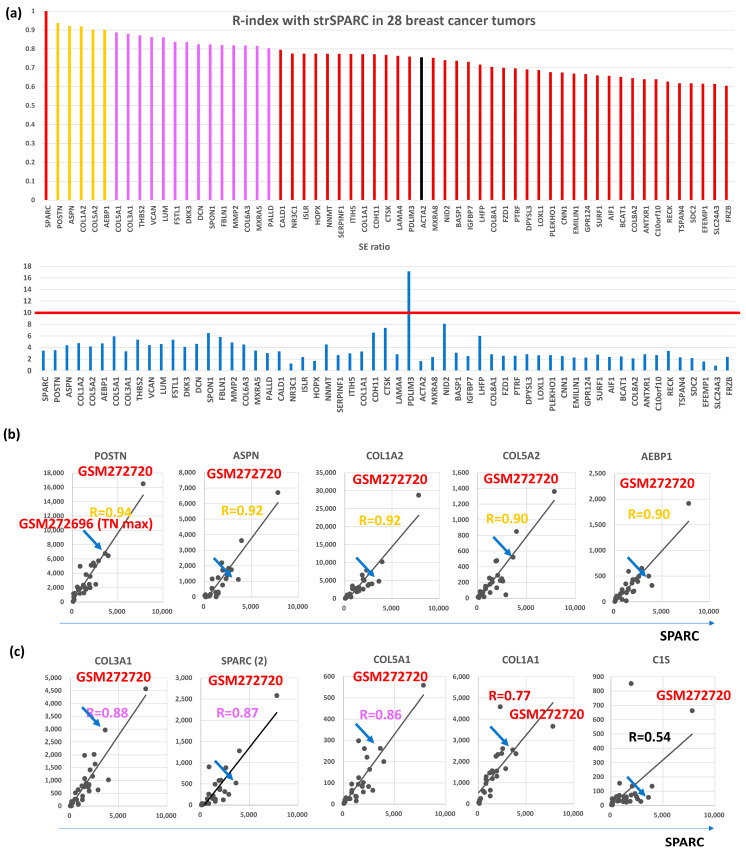
Correlation with SPARC in cancer stroma and S/E ratio of 28 breast cancer cases in GSE10797. (**a**) Upper panel, the correlation coefficient with SPARC was calculated in the stroma of 28 breast cancer cases, and 58 genes with R ≥ 0.6 (59 genes including SPARC) were finally selected. Gold bars represent R ≥ 0.9, purple bars represent R ≥ 0.8, and reddish brown bars represent R ≥ 0.6, and black bars is CAFs markers (ACTA2 in the figures). Lower panel shows SE ratio of the above 59 genes. The red line is drawn at 10 because an SE ratio of 10 or beyond is considered a CAFGs. (**b**) All 5 genes associated with SPARC expression in cancer stroma of the BC tumors (R ≥ 0.9). The highest expression was reproducibly confirmed in GSM272720, while the blue arrow shows the BC case with the highest expression among Triple negative (TN) type BC (TN max). (**c**) Representative SPARC target genes identified by SPARC knockdown in YS1 cells (CAFs cell line), in which their expression was associated with SPARC expression in cancer stroma. The figure includes another SPARC probe (SPARC(2)), which had inconsistent data.

**Figure 2 biomedicines-12-01964-f002:**
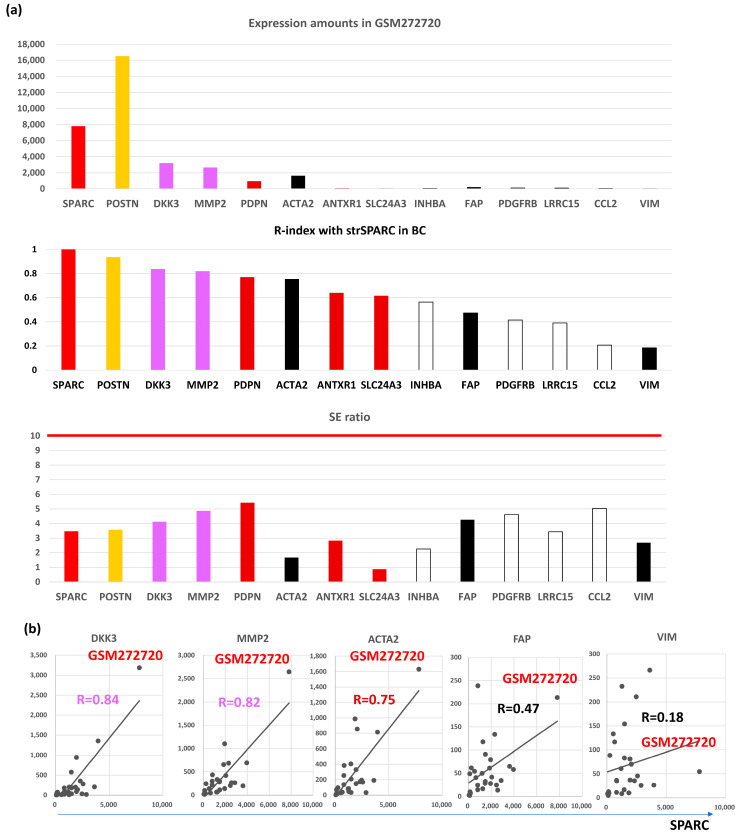
Molecular features of representative CAFGs in BC (BC CAFGs) among CRC CAFGs. (**a**) Upper panel, expression amounts of CAFGs in GEM272720. Middle panel, correlation coefficient between individual CAFGs and strSPARC in BC. Lower panel, SE ratio (all below 10). (**b**) Correlation of BC CAGs (DKK3 and MMP2) and well-known CRC CAFGs (ACTA2, FAP, and VIM) to strSPARC in BC is shown, in which FAP and VIM are not strongly associated with strSPARC (R = 0.47 and R = 0.18, respectively).

**Figure 3 biomedicines-12-01964-f003:**
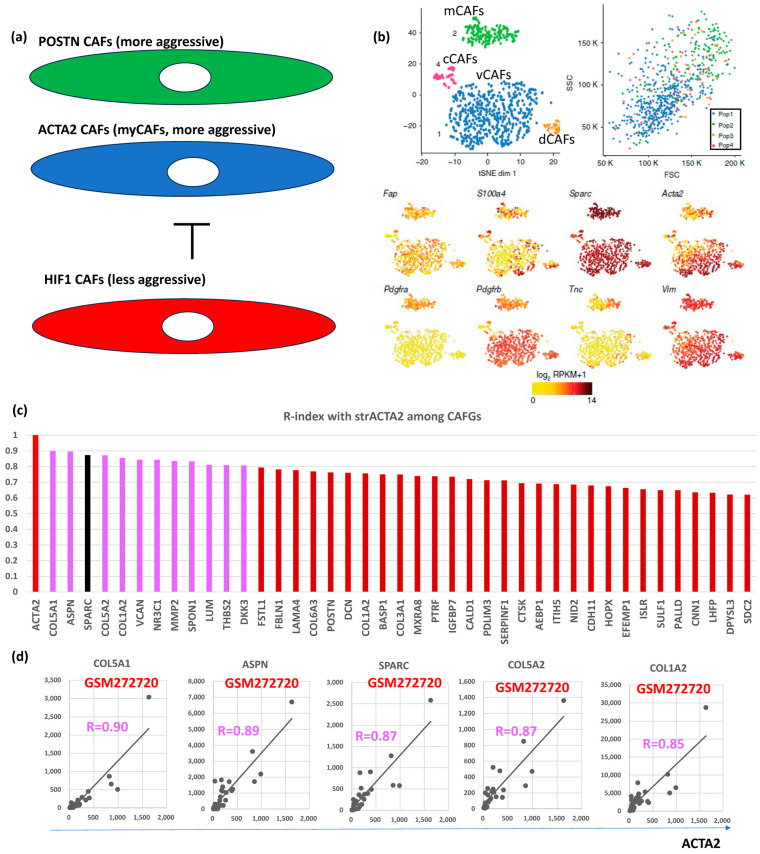
BC CAFGs and current understanding its functional relevance. (**a**) Relationship between CAFs expressing POSTN and/or ACTA2 (myCAFs) and CAFs expressing HIF1A. (**b**) scRNA analysis was performed on mesenchymal cells from mouse breast cancer and showed that they could be classified into 4 CAFs categories (vascular CAFs-vCAFs, matrix CAFs-mCAFs, cycling-cCAFs, and developmental CAFs-dCAFs). Well known CAFs markers (FAP, SPARC, ACTA2, and VIM) were expressed in all 4 CAFs, while PDGFRA was expressed only in mCAFs (this figures was made from reference [3]). (**c**) Correlation coefficient (R = 0.6 or beyond) between CRC CAFGs and strACTA2 in BC. (**d**) Representative correlation of CRC CAFGs associated with ACTA2 expression in cancer stroma.

**Figure 4 biomedicines-12-01964-f004:**
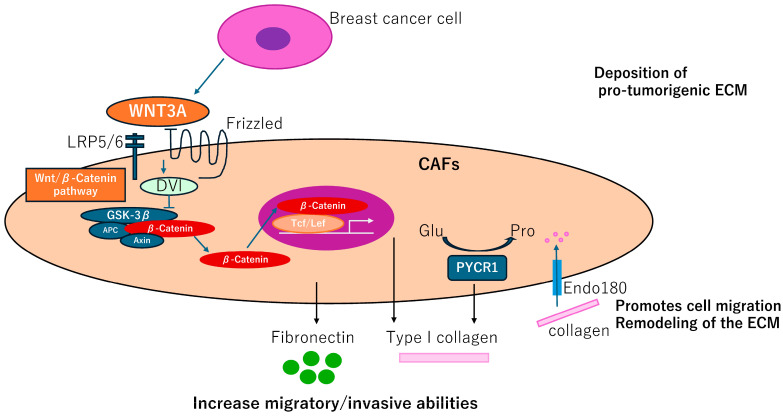
CAFGs collagens in BC and novel therapeutic potential. Adipocyte-derived fibroblasts adjacent to breast tumor cells increase secretion of fibronectin and collagen I, promoting migration and invasive potentials, which can be dependent on activation of the Wnt/β-catenin pathway. Endo180 promotes cell migration and ECM remodeling by collagen uptake. PYCR1 is a key enzyme in proline synthesis and is involved in the production of ECM that promotes tumorigenesis and may be a therapeutic target.

**Figure 5 biomedicines-12-01964-f005:**
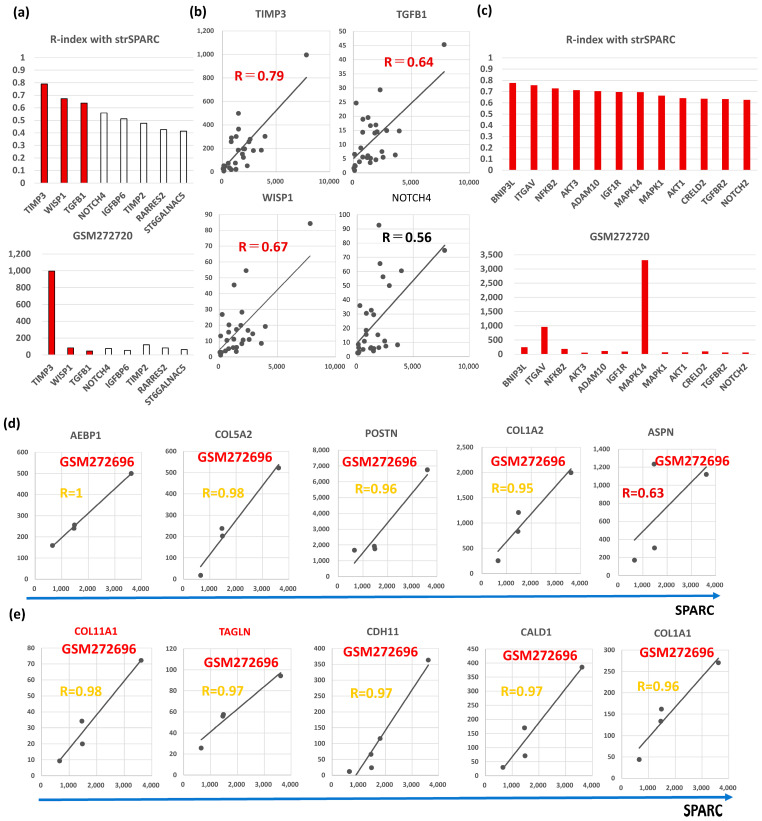
Molecular characteristics of Semi-CAFGs in BC that may be involved in CAFs activation and have novel therapeutic potential. (**a**) Upper panel, correlation coefficients (R-index) between genes defined as Semi-CAFs in CRC and strSPARC. R-index = 0.6 or beyond represents colored bars. Lower panel, expression amounts of each Semi-CAFs in GSM272720. Outstandingly, TIMP3 expression was robust in cancer stroma of the BC tumors. (**b**) Semi-CAFGs (TIMP3, WISP1, TGFB1, and NOTCH4) expressions associated with SPARC expression in cancer stroma. (**c**) Upper panel, correlation coefficients (R-index = 0.6 or beyond with strSPARC) in BC among CRC CAFGs. Lower panel, expression amounts of each L-CAFs in GSM272720. (**d**) Correlation of CAFGs with strSPARC in BC (R ≥ 0.9) commonly in whole BC and TNBC was individually shown. (**e**) Correlation of CAFGs (COL11A1 and TAGLN) with strSPARC (R ≥ 0.9) only in TNBC was individually shown.TIMP (tissue inhibitor of metalloproteinase) is an inhibitor of the matrix metalloproteinase (MMP) family. Quadruple TIMP knockout (TIMPless, *TIMP1*, bSE = 1.7; *TIMP2*, bSE = 2.6; *TIMP3*, bSE = 3.1; *TIMP4*, bSE = 2.0) fibroblasts to unleash MMP activity and show that complete Timp loss is sufficient for the acquisition of hallmark CAFs functions in BC [40]. These findings suggested that MMP activity is critical for CAFs activity in BC.

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
