# Peer review of "CAF-Associated Genes in Breast Cancer for Novel Therapeutic Strategies"

_biomedicines, 2024, doi:10.3390/biomedicines12091964_

Round 1

Reviewer 1 Report

Comments and Suggestions for Authors

The authors conducted a systematic review on the utilization of CAFS-related genes in breast cancer treatment. They compared the molecular characteristics of CAFs in breast cancer with colorectal cancer and identified distinct molecular features specific to breast cancer-associated CAFs.

The molecular characteristics of BC CAFs, the associated functions of BC CAFG, and its potential in the field of innovative therapies were described. Furthermore, the therapeutic potential of Semi-CAFGs and low SE CAFGs (L-CAFGs) in BC was further discussed.

The current form of this review can be accepted due to its exceptional quality, characterized by a comprehensive literature summary and in-depth discussion. Moreover, the language used is exemplary. If desired by authors, it would be appropriate to provide a brief description of BC treatment in the background section. Additionally, more relevant literature can be appropriately cited in this section.

Author Response

The authors conducted a systematic review on the utilization of CAFS-related genes in breast cancer treatment. They compared the molecular characteristics of CAFs in breast cancer with colorectal cancer and identified distinct molecular features specific to breast cancer-associated CAFs.

The molecular characteristics of BC CAFs, the associated functions of BC CAFG, and its potential in the field of innovative therapies were described. Furthermore, the therapeutic potential of Semi-CAFGs and low SE CAFGs (L-CAFGs) in BC was further discussed.

The current form of this review can be accepted due to its exceptional quality, characterized by a comprehensive literature summary and in-depth discussion. Moreover, the language used is exemplary. If desired by authors, it would be appropriate to provide a brief description of BC treatment in the background section. Additionally, more relevant literature can be appropriately cited in this section.

→Thank you for your pertinent comments. I actually inserted the following sentence in the first paragraph, the second sentence of the "Introduction" on page 1.

Drug therapy for BC depends on the molecular subtypes: endocrine therapy for luminal BC, anti-HER2 drugs for HER2 BC, and chemotherapy and immune checkpoint inhibitors (ICIs) for TNBC.

Reviewer 2 Report

Comments and Suggestions for Authors

In this review article, molecular features of cancer-associated fibroblasts (CAFs) in breast cancer (BC) in comparison to colorectal cancer (CRC) were summarized, and unique molecular features of BC CAFs was identified. This manuscript may be accepted for publication after minor revisions.

1, It is very high percent match “39%” based on iThenticate report. Please carefully revise the manuscript to lower the percent match.

2, The abbreviations can be deleted from the title, subtitle, and abstract. For example, the (CAFGs)” can be removed from the title. The “(BC CAFGs)” can be deleted from the subtitle “2. CAFGs in BC (BC CAFGs)”.

3, The introduction section can be shortened, and most of the content in the introduction section (current version) can be put in a separated section.

4, The future prospective can be emphasized in the conclusion section.

5, The English language can be further improved. Please avoid using first-person narratives.

6, The quality (resolution) of figures can be improved.

Comments on the Quality of English Language

1, It is very high percent match “39%” based on iThenticate report. Please carefully revise the manuscript to lower the percent match.

2, Please avoid using first-person narratives.

Author Response

In this review article, molecular features of cancer-associated fibroblasts (CAFs) in breast cancer (BC) in comparison to colorectal cancer (CRC) were summarized, and unique molecular features of BC CAFs was identified. This manuscript may be accepted for publication after minor revisions.

1, It is very high percent match “39%” based on iThenticate report. Please carefully revise the manuscript to lower the percent match.

→ Thank you for your comments. I rewrote the original description to our way to describe each issue.

2, The abbreviations can be deleted from the title, subtitle, and abstract. For example, the “(CAFGs)” can be removed from the title. The “(BC CAFGs)” can be deleted from the subtitle “2. CAFGs in BC (BC CAFGs)”.

→ Thank you for pointing this out. I agree this comment. I deleted the abbreviations from the title, subtitle, and abstract throughout this paper.

3, The introduction section can be shortened, and most of the content in the introduction section (current version) can be put in a separated section.

Thank you for your suggestion.  Accordingly, I divided the original introduction chapter was changed into separate chapters in the revised ones.

4, The future prospective can be emphasized in the conclusion section.

→Thank you for pointing this out. I agree this comment. I added the following sentence.

Limitations of this study include the small sample size of the data analyzed, especially for TNBC.  The data used in this study may not be representative of the broader population of BC patients, and further studies using larger datasets and different patient populations are needed to validate the findings regarding BC relevance of ACTA2.  Moreover, the greatest limitation of this study is that it relied exclusively on gene expression data to identify CAF markers and did not authenticate the findings through other methodology such as functional assays or protein expression, given that this paper is a kind of review article. 

Recent studies clarified future prospective that TME shows a hierarchy of cell-cell interactions dominated by CAFs {Mayer, 2023 #41}, where Mayer S et al provided an experimental-mathematical approach to decompose the TME into specific small circuits of interacting cell types.  They herein discovered, using a scRNAseq data, a hierarchical network of interactions, with CAFs at the top secreting factors primarily to TAMs, and actually identified a mediator of CAFs-TAMs interactions. The complexity of the TME may be thus simplified by identifying such small circuits, facilitating the development of novel strategies to modulate the TME.

5, The English language can be further improved. Please avoid using first-person narratives.

→  Thank you for pointing this out. I agree this comment. I have modified them as follows.

  • Abstract 3rd sentence of the page1

In a recent study, 115 CAFs-associated genes (CAFGs) were identified in a public database of

microdissection and microarray data (GSE35602) from 13 colorectal cancer (CRC) tumors.

  • 2, CAFs classification and heterogeneity in BC first sentence of the fourth paragraph

Recently, CAFs-associated genes (CAFGs) in colorectal cancer (CRC) were defined based on two criteria: (1) strong association (R ≥ 0.9, as indicated by underlined genes throughout this study) with SPARC expression, a well-established stromal marker [7], in the cancer stroma of CRC tumors (GSE35607), and (2) specificity to the stroma (stroma/epithelia expression ratio, SE ratio ≥ 10) similar to SPARC (SE = 17.2).

  • 2, CAFs classification and heterogeneity in BC second sentence of the sixth paragraph

Subsequently, in silico analysis was conducted on data from 28 micro-dissected breast cancer tissues (GSE10797) [5], similar to those of colorectal cancer, in order to comprehend and summarize CAFGs in breast cancer.

  • 4, BC CAFGs and current understanding its functional relevance the last sentence of the first paragraph

Recently, SPARC-target genes (COL3A1, COL5A1, COL1A1, and C1S) were identified through SPARC knockdown in YS1 cells (a CAF cell line, data not shown), and these genes similarly showed close association with SPARC in breast cancer stroma, as depicted in Fig. 1c.

  • Conclusion the first sentence of the second paragraph

This review article summarizes the molecular features of cancer-associated fibroblasts (CAFs) in breast cancer compared to colorectal cancer (CRC), and identifies the unique molecular features of CAFs in breast cancer.

  • Conclusion  the fifth sentence of the second paragraph

Moreover, among breast cancer CAFGs, POSTN expression was found to be the highest, followed by DKK3 and MMP2.

6, The quality (resolution) of figures can be improved.

→ Thank you for pointing this out. I agree this comment.  The resolution was improved by increasing dpi of the original files. 

Reviewer 3 Report

Comments and Suggestions for Authors

The article titled " CAFs-associated genes (CAFGs) in breast cancer for novel therapeutic strategy by Kanako Naito et al. provides valuable insights. However, there are several areas that require attention:

1. In the abstract, The statement that breast cancer is a genetic disease is not entirely accurate.

2. The data used in the study may not be representative of the broader population of breast cancer patients. Further studies using larger datasets and different patient populations are needed to validate the findings

3. The study's sample size is fairly small, which could limit the generalizability of the results

4. While the study's results suggest that ACTA2 may be the most relevant CAF marker in breast cancer, it is important to note that the study only analyzed a small number of breast cancer tumors

5. One limitation of the study is that it relied exclusively on gene expression data to identify CAF markers and did not authenticate the findings through other methodology such as functional assays or protein expression

6. Authors did not address the potential heterogeneity of CAFs within breast tumors. It is possible that diverse subpopulations of CAFs with different molecular features and functions may exist within the same tumor, which could have significant implications for therapy development.

Comments on the Quality of English Language

Moderate editing of English language required

Author Response

The article titled " CAFs-associated genes (CAFGs) in breast cancer for novel therapeutic strategy” by Kanako Naito et al. provides valuable insights. However, there are several areas that require attention:

  1. In the abstract, The statement that breast cancer is a genetic disease is not entirely accurate.

→Thank you for pointing this out. Sorry for my misunderstanding. Therefore, I have modified the first paragraph of the “Introduction”.

  1. The data used in the study may not be representative of the broader population of breast cancer patients. Further studies using larger datasets and different patient populations are needed to validate the findings

GSE10797 includes 28 BC patients, and as a study handling the microdissected tumors the number is considered to be high.  For example, GSE3602 handling CRC included only 13 cases.  Nevertheless, your suggestion is important concern.  I added this debates to the limitation of this data analysis.

  1. The study's sample size is fairly small, which could limit the generalizability of the results

You may describe TNBC cases. I described your concerns in the Limitation.

  1. While the study's results suggest that ACTA2 may be the most relevant CAF marker in breast cancer, it is important to note that the study only analyzed a small number of breast cancer tumors

→ Since the best data available to us and most suitable for analysis is GSE10797 used in this study, it is difficult to increase it further. I described your concerns in the Limitation.

  1. One limitation of the study is that it relied exclusively on gene expression data to identify CAF markers and did not authenticate the findings through other methodology such as functional assays or protein expression

→This study is basically a review paper, however in order to comprehensive understanding of CAFs-associated genes, BC data was utilized like a research study, however as it is a review article, functional study was not done.  However anyway we put the methodology description in the text on page, line. 

  1. Authors did not address the potential heterogeneity of CAFs within breast tumors. It is possible that diverse subpopulations of CAFs with different molecular features and functions may exist within the same tumor, which could have significant implications for therapy development.

→ We described 2 ways of heterogeneity of BC CAFs as following of the original paper.  One is vCAFs, mCAFs, cCAFs, and dCAFs in Fig. 2, and another one is myCAFs and iCAFs.  In cancers other than BC, apCAFs and POSTN CAFs may be independent of the classification, however they have not be definitive right now in BC, and should not be included in this review paper.

Reviewer 4 Report

Comments and Suggestions for Authors

Comments:

1. The write-up and sentence structure should be improved in abstract section as it is the most important part of any article e.g. “Breast cancer (BC) is the most common cancer in women and is a genetic disease" should be modified to improve its scientific sense. Furthermore, BC is also influenced by other factors other than genetic setup.

2. Authors are suggested to provide more explanation to explore further the impact of mutations (PI3K, TP53, BRCA1/2) on tumor behavior and treatment response

3. Authors mentioned the identification of different CAF subtypes using scRNA technology. Authors should provide their specific roles in BC progression and therapy resistance with references to primary studies.

4. Authors must provide the methods used for data extraction and analysis including the bioinformatics tools and statistical tests applied.

5. Authors should provide the functional implications of SPARC's unique association
with COL11A1 and TAGLN in TNBC?

6. Authors are suggested to provide the importance of SE ratios.

7. Authors are suggested to discuss the contribution of identified CAFGs to BC progression.

8. Authors are suggested to discuss the functions of different CAF subtypes and link them with the pathways they influence in BC.

9. Authors should provide the future directions.

10. Authors are suggested to enlist the limitations of the study.

11. Authors should justify the similarity index (39%) and its should be within acceptable limit (less than 20%).

Comments on the Quality of English Language

Improve the sentence structure throughout the article.

Author Response

  1. The write-up and sentence structure should be improved in abstract section as it is the most important part of any article e.g. “Breast cancer (BC) is the most common cancer in women and is a genetic disease" should be modified to improve its scientific sense. Furthermore, BC is also influenced by other factors other than genetic setup.

→Thank you for pointing this out. Sorry for my misunderstanding. Therefore, I have modified the first paragraph of the “Introduction”.

  1. Authors are suggested to provide more explanation to explore further the impact of mutations (PI3K, TP53, BRCA1/2) on tumor behavior and treatment response

→ Thank you for pointing this out. I added explanations describing the mutations.

PI3K regulates signaling pathways important for cell growth, survival, and metabolism; overactivation or abnormal mutations in PI3K can promote cell growth and cause resistance to treatment in BC {Di Cosimo, 2010 #117}, which can make it difficult for patients with certain PI3K mutations to respond to certain therapies.TP53 is a tumor suppressor gene that detects and repairs cellular DNA damages or abnormalities or induces cell death (apoptosis) {Tanaka, 2018 #118}. Mutations in TP53 can impair the ability of cells to repair DNA and increase cancer cell progression and resistance to therapy. Therefore, TP53 mutations may play an important role as a predictor of therapeutic resistance.

BRCA1 and BRCA2 are genes involved in DNA repair, and mutations in both genes reduce the ability of DNA double-strand break repair. BC patients with BRCA1/2 mutations are known to be highly sensitive to PARP inhibitors, especially Olaparib {Fong, 2009 #119}. This makes BRCA1/2 mutations a potentially important predictor in treatment selection. Such genetic mutations may also affect the individual pathology and prognosis of BC. Therefore, the presence or absence of these mutations is an important factor to consider in the choice of treatment strategies and prognostic prediction.

  1. Authors mentioned the identification of different CAF subtypes using scRNA technology. Authors should provide their specific roles in BC progression and therapy resistance with references to primary studies.

→Thank you for pointing this out. I additionally described roles of the different CAFs subtypes in BC progression and therapy desistance on page, line. 

In the primary study, signatures from vCAFs and mCAFs have demonstrated prognostic capabilities by correlating with metastatic dissemination in two large clinical cohorts involving over 2600 breast cancer patients. Additionally, the mCAF signature showed correlation with a stromal signature predictive of treatment response. These findings collectively present an enhanced understanding of breast cancer CAFs at cellular, molecular, and functional levels, suggesting potential for developing targeted drugs or biomarkers of clinical significance with improved precision.

  1. Authors must provide the methods used for data extraction and analysis including the bioinformatics tools and statistical tests applied.

→Thank you for pointing this out. Methods were newly added in this paper. 

  1. Authors should provide the functional implications of SPARC's unique association
    with COL11A1 and TAGLN in TNBC?

→Other subtypes of breast cancer are predominantly i-CAF, and it is suggested that myCAF may be stronger in TNBC, but due to the small number of cases, we cannot mention that much at this time, and therefore, I described it in the Limitation.

  1. Authors are suggested to provide the importance of SE ratios.

→Thank you for pointing this out. It is explained in chapter “2, CAFs classification and their heterogeneity in BC“.

  1. Authors are suggested to discuss the contribution of identified CAFGs to BC progression.

→Thank you for pointing this out.  I have stated in chapter “4. BC CAFGs and current understanding its functional relevance”.

  1. Authors are suggested to discuss the functions of different CAF subtypes and link them with the pathways they influence in BC.

→Thank you for pointing this out. It is explained in the “CAFs classification and heterogeneity in BC “ chapter.

  1. Authors should provide the future directions.

→Thank you for pointing this out. I agree this comment. I would add the future directions.

  1. Authors are suggested to enlist the limitations of the study.

→Thank you for pointing this out. I agree this comment. I would add the limitations of the study.

Limitations of this study include the small sample size of the data analyzed, especially for TNBC.  The data used in this study may not be representative of the broader population of BC patients, and further studies using larger datasets and different patient populations are needed to validate the findings regarding BC relevance of ACTA2.  Moreover, the greatest limitation of this study is that it relied exclusively on gene expression data to identify CAF markers and did not authenticate the findings through other methodology such as functional assays or protein expression, given that this paper is a kind of review article. 

Recent studies clarified future prospective that TME shows a hierarchy of cell-cell interactions dominated by CAFs {Mayer, 2023 #41}, where Mayer S et al provided an experimental-mathematical approach to decompose the TME into specific small circuits of interacting cell types.  They herein discovered, using a scRNAseq data, a hierarchical network of interactions, with CAFs at the top secreting factors primarily to TAMs, and actually identified a mediator of CAFs-TAMs interactions. The complexity of the TME may be thus simplified by identifying such small circuits, facilitating the development of novel strategies to modulate the TME.

  1. Authors should justify the similarity index (39%) and its should be within acceptable limit (less than 20%).

→ Thank you for pointing this out. I would change the sentence.

Round 2

Reviewer 3 Report

Comments and Suggestions for Authors

Accept in present form

Comments on the Quality of English Language

 Moderate editing of English language required.

Reviewer 4 Report

Comments and Suggestions for Authors

Authors addressed all the comments and suggested changes have been incorporated. The article has been improved. However, authors must re-evaluate the content of the article with respect to the similarity index. It should be falls within the accepted limit (below 20%). At the moment, according to The iThenticate document viewer, it is 35%.